# Immunology Meets Artificial Intelligence: Expanding Our Scientific Toolbox

## Abstract

Artificial intelligence (AI) is now a part of our daily lives. In this swiftly evolving landscape, AI has become an indispensable tool in the scientific discovery process, augmenting tasks from ideation and hypothesis generation to data cleaning, code development and debugging, text editing, and data analysis. This paper advocates for educational resources for AI in immunology, emphasizing its unique position to leverage AI's potential for scientific discovery. Immunology's intricate tapestry spans multiple biological scales, from molecular interactions to complex systems, presenting an ideal canvas for AI-driven solutions. The field is rich in data, thanks to advanced molecular and single-cell technologies, making it ripe for AI-driven insights. To support the intersection of AI and immunology, we've established a dedicated website as an AI resource hub, offering curated modules and resources. By fostering a "learn by playing" ethos, promoting interactive and engaging workshops, and inviting community contributions, we aim to empower immunologists to harness AI's transformative capabilities and navigate this exciting frontier collectively.

## 1   Introduction

Artificial intelligence (AI) has rapidly integrated into our daily lives, moving from the realm of science fiction to an omnipresent reality. A prime example of this phenomenon is the remarkable ascent of AI chatbots like ChatGPT, which reached 100 million households within a few months—a feat that traditional landline telephones took 75 years to achieve [1-4]. Furthermore, the proliferation of AI tools in the commercial market shows no signs of slowing, with an incessant influx totalling thousands of new tools and an expected AI market size of $407 billion by 2027 [5,6].

In this swiftly evolving landscape, AI has become an indispensable tool in the scientific discovery process, augmenting tasks from ideation and hypothesis generation to data cleaning, code development and debugging, text editing, and data analysis [7-9]. The question transcends academic disciplines, leaving every scientific field wrestling with how to effectively incorporate AI tools into research practices, educational programs, and to provide comprehensive technical training for students, faculty, and staff [7,10].

Consequently, there is a pressing need to bolster educational resources as gateways to bridge the ever-widening computational literacy gap. While fields like computational biology [11], genomics [12, 13], and cheminformatics [14,15] have made substantial strides in establishing robust computing frameworks over recent decades, one scientific domain stands out as uniquely positioned for transformation through AI: immunology [16-21]

Submitted to NeurIPS 2023 AI for Science Workshop.

We recognized the vastness of intellectual real estate in immunology and the overwhelming speed of AI advancement. Because of this pressure to keep up with AI tools and research, we created the AI for Immunology website ("AI 4 Immuno") to pool and disseminate resources for the community. In sum, this paper champions the union of AI and immunology as an essential and pioneering partnership that can bolster the computational immunology toolkit.

## 1.1 Why AI for immunology?

**Immunology is uniquely poised for AI acceleration.**   Immunology is everywhere. As a discipline, this domain is an expansive and intricate tapestry that spans multiple biological scales. From protein expression, cytokine signaling, and single-cell interactions all the way up to tissue organization across multiple organs in complex systems. Immunology is uniquely positioned for transformation through AI because of its inherent complexity and richness.

**Complex knowledge spanning multiple scales of biology.**   One striking aspect of immunology is its sheer breadth and complexity. It is virtually impossible for a single immunologist to become an expert in every model system, immune cell type, immune-related disease setting from cancer to autoimmunity, pathogen type (encompassing viruses, parasites, bacteria, fungi), and to comprehend the diverse ways these diseases manifest as symptoms across dozens of phenotypes [22].

**The field is rich! (in data and knowledge).**   Immunology stands out as a discipline uniquely enriched by vast repositories of big data, thanks to a multitude of advanced molecular and next-generation single-cell technologies [18,19,23, 24]. Immunologists employ a diverse array of bench techniques (molecular and cellular) to interrogate biological and disease interactions at high resolution, from cutting-edge gene editing to multi-generational breeding and cell line engineering [25,26].

These cutting-edge tools, such as various flavors of cytometry, enable researchers to dissect the intricacies of the immune system at unprecedented levels of granularity [27-30]. In addition, new technologies, like multi-color imaging techniques and high-throughput sequencing, each demand their own preprocessing pipelines [31-34].

The rich tapestry of data extends from the molecular interactions within individual cells to the broader systems-level understanding of immune responses across various tissues and organs. This immense data landscape, brimming with intricate biological details, makes immunology an ideal candidate for AI-driven solutions to decipher complex patterns, extract meaningful insights, and accelerate scientific discovery [35].

## 2 Creating an AI resource hub for the field

Immunology experts possess a profound grasp of existing gaps, technical data limitations, biological variations, and research challenges. We hope this resource hub makes it easier for scientists to navigate the AI landscape for their own domain-specific applications. The scale of both challenges and potential applications in immunology is growing exponentially, positioning it as the frontier where AI's capabilities can shine.

In a rapidly evolving landscape, staying informed about the latest AI developments can be over-whelming. With a multitude of AI news and tools available online, it can be challenging to discern what truly works for your needs and what might be a gimmick.

## 2.1 Our approach

To address this, we have created a dedicated website that serves as a curated AI resource hub, called "AI for Immunology" *(URL redacted per anonymity guidelines)*. Here, immunologists and researchers can find valuable information, featuring learning modules and supplementary educational resources, all with the overarching goal of preparing future immunologists to lead in the development of computational immunology infrastructure.

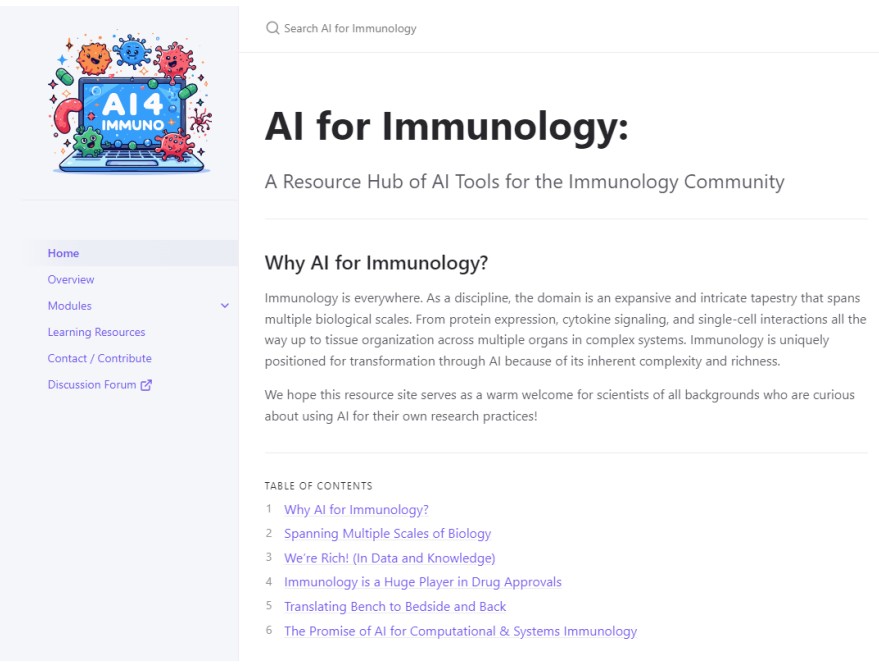

Figure 1: The landing page for the AI for Immunology website.

Our aim is to provide a reliable platform where the immunology community can navigate the AI landscape with confidence, ensuring that they harness the most beneficial tools to advance their research and computational capabilities.

## 2.2 Streamlined "AI 4 Immuno" web interface

The resource hub we created is available online at *<redacted URL per anonymity guidelines>*. The website is optimized for viewing on desktop and mobile devices.

## 3 Module Structure

The website's module structure is divided into several categories:

- Module 01 - Overview of the Generative Application Landscape
- Module 02 - Curated AI Academic Research Tools
- Module 03 - Using AI Chatbots in Research
- Module 04 - Pair Programming with AI-Powered Tools
- Module 05 - Assisting with Analysis Workflows
- Module 06 - Building Interactive Tools
- Module 07 - Creating Generative Art
- Module 08 - Applications of AI Tools in Immunology

To illustrate the real-world impact, we also present examples drawn from various technology blogs and social media platforms, showcasing how general users, often with minimal prior programming knowledge, have harnessed AI tools like ChatGPT and GitHub Copilot to create websites, whole applications, and software demos.

## 3.1 Generative application landscape

The generative AI landscape has exploded globally over the past couple of years. It is impossible to calculate the exact number of total AI applications on the market, but some sites estimate that there are currently over 18,500 AI startups in the United States focused on deploying AI-based tools, software, and services across a wide range of sectors [36-38].

Within this module, we provide a high-level perspective of current AI tools on the market. In addition, we outline the differences between and history of artificial intelligence, machine learning, and deep learning. We decided to keep our explanations brief on AI models for research and software development considering the expansive body of work from the computational research field.

## 3.2 Curated AI academic research tools

For academic researchers, sifting through the overwhelming amount of research can be a challenge. To help with this, there are AI tools tailored to their needs. We have provided a brief list of the most relevant AI tools for the academic community. These tools are invaluable in a world where academic knowledge is growing rapidly, making it easier for researchers to access, understand, and stay updated in their fields.

In this module, visitors can read about a handful of multipurpose academic research tools. First, we cover general AI chatbots (ChatGPT, Bard, and Bing Chat) which provide versatile assistance on various tasks, from answering questions to offering research guidance [39]. We then proceed into specialized chatbots designed for academia, which help researchers with literature searches, organizing papers, and managing citations, making research workflows smoother. We also cover a few AI tools which assist in the literature review process, such as quickly scanning and summarizing large amounts of text, ultimately saving researchers time and effort. Finally, we highlight a tool which can help researchers gauge consensus from an academic community on common themes, controversies, and emerging trends in research.

## 3.3 Using AI chatbots in research

For this module, we discuss the differences in AI chatbot performance and outputs when given the same prompt. We also share resources for scientists hoping to better understand and effectively utilize prompt engineering principles to support their research needs.

## 3.4 Pair programming with AI-powered tools

There are two new modes of programming: no code development [40] and low code (also called AI-assisted) development [41-43]. The former provides interactive interfaces which allow users to build websites and applications without writing any code, while the latter incorporates features such as predictive code completion and active debugging assistance. Within this module, we cover several ways in which immunologists can integrate AI into their programming workflow. We highlight the distinction between interacting with AI chatbots within internet browsers compared to AI-assisted pair programming within interactive development environments (IDE).

## 3.5 Assisting in analysis workflows

In this module, we focus on how AI can support streamlining and simplifying the various stages of data analysis workflows [44]. Standard preprocessing steps, often time-consuming, are addressed, including data cleaning, parsing, and creating high-level overviews. We provide resources which offer guidance on breaking down complex tasks into manageable steps, constructing effective workflows and roadmaps, and seamlessly connecting individual steps into a standard preprocessing pipeline. All of the above provide immunologists with practical considerations for enhancing their data analysis processes.

### 3.6 Building interactive tools

Within this module, we explore various ways in which immunologists can create interactive resources. This includes building general project landing pages, browser applications, and online documentation/manuals. We provide insights into templating, and constructing both front and back-end programs via generative coding. Additionally, we delve into the realm of interactive web-based agents, such as chatbots built into knowledge graphs, offering a look at how these personalized tools can enhance the immunology toolkit.

### 3.7 Creating generative art

The art world is currently reckoning with generative AI [45,46]]. While some individuals consider AI to replace human creators altogether, many others think of AI as an additional tool which aids in creation [47,48]. There are many concerns about the training and/or input data used to power these models and whether creators' consent is appropriately taken into consideration [49-52].

In this module, we cover potential use cases for generative art in science, such as personalized lab logos, graphical abstracts for research papers, brainstorming potential journal covers, creating conceptual icons for slideshows or badges and stickers for your projects. We also taper expectations about generative art tools in science by showing an example using the same prompt, which resulted in very different design outcomes. The current reality is that these tools cannot directly replace the talent and expertise of trained scientific and medical illustrators.

### 3.8 Potential Applications in Immunology

In this module, we feature examples of AI-powered applications in the field, such as an application where researchers trained an immunology knowledge assistant or generated an interactive web dashboard. We are actively gathering prime examples of AI-assisted development in immunology. Our aim is to keep highlighting incredible projects at this exciting intersection.

## 4 Sharing resources for further exploration

### 4.1 Staying up-to-date

Although the most cutting-edge research is published in premier academic journals, announcements for AI tools are found in a host of other online places. In most cases, advances in AI tools are not written up as research papers and peer-reviewed.

Instead, the latest news can be found on social media platforms, community forums, professional networking sites, personal blogs, and technology media sites. We provide a page featuring dozens of recommendations, including scientific workshops and annual conferences outside of immunology as well as books covering emerging technology trends. We plan to update this list of resources over time based on suggestions received from the community.

### 4.2 Formal schooling vs. self-taught methods

In today's ever-evolving landscape of knowledge, the traditional model of returning to school for new degrees in emerging domains may not always be feasible. Instead, the role of scientists is increasingly that of lifelong learners.

Massive open online courses (MOOCs) have emerged as a powerful tool to facilitate this continuous learning journey. Over the past decade, MOOCs have witnessed a remarkable increase in quality, credibility, and popularity for STEM education [53-57]. Academic institutions and industry leaders have collaborated to offer MOOCs across a wide range of subjects, from coding to data science, prompt engineering for both beginners and seasoned developers, stable diffusion, and more. On the resources page, we provide a brief list of MOOCs to give scientists an idea of the variety of online self-study courses available.

## 5  Welcoming community contributions

We hope the AI for Immunology resource hub fosters a collaborative spirit. Input from the immunology community is invaluable to us and we welcome folks to share their thoughts, ideas, recommendations, and constructive criticism! Additionally, we are in the process of creating a content submission form where scientists can suggest new resources, papers, or case studies to be featured on the website. We eagerly encourage members of the community to provide feedback and suggestions to make the website even more practical and user-friendly. Our hope is that this collaborative approach will lead to an enriched resource hub that resonates with and empowers the immunology community.

## 6  Challenges

**Translating bench to bedside and back.**  Immunology's prominence in the medical landscape is unmistakable, with a significant portion of FDA drug approvals in 2022 driven by the field of immunology [58-60]. The scale of both challenges and potential applications in immunology is growing exponentially, positioning it as the frontier where AI's capabilities can shine.

However, this endeavor is not without its challenges. Basic researchers and clinicians sometimes employ the same terminology, albeit coded with different meanings. The complexities of translating clinical observations into research questions (and vice versa), navigating the vast diversity of clinical phenotypes, and bridging disciplinary languages all pose significant hurdles [61,62].

## 7  Conclusions and Future Directions

In conclusion, this paper highlights the transformative potential of AI in the field of immunology. It underscores the urgency of integrating AI-powered tools and providing educational resources directly to immunologists amidst the ever-expanding AI landscape. We are excited to offer a dedicated website, as an AI resource hub, filled with curated examples and helpful references.

There are several promising directions to explore in the future. To further support the community, we aim to expand the website's resources, linking more opportunities for developing data science and coding skills and organizing more detailed examples gathered from the field.

We advocate for a shift towards a "learn by playing" ethos, which can make learning about AI tools less intimidating and more accessible and engaging. Moreover, we hope this work inspires hands-on workshops at immunology venues, fostering knowledge sharing and collaboration among researchers.

As a community-driven endeavor, we invite contributions from researchers throughout the field to continually update the website, recognize and celebrate applications of AI tools in immunology, and collectively learn to navigate the new territory of artificial intelligence in immunology. We are excited to build a vibrant platform that fosters excitement, collaboration, and engagement among researchers in this dynamic field.

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

## Appendix

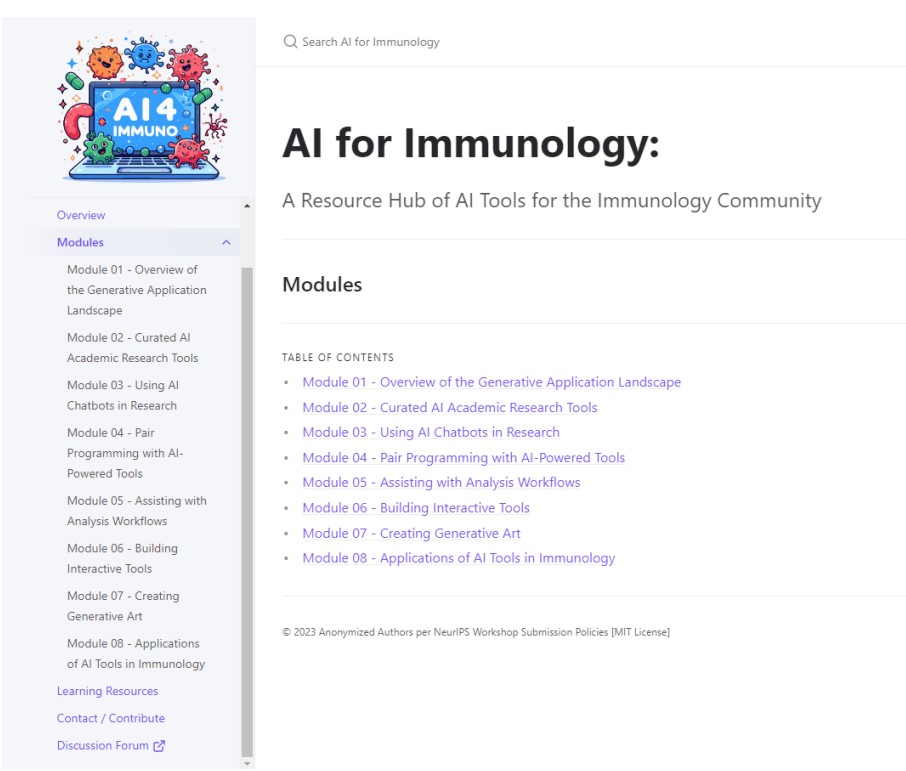

Figure 2: The module navigation page.

Figure 3: Page for module 01.

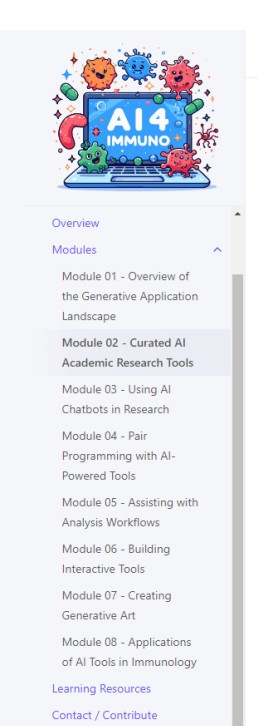

Q Search AI for Immunology

Modules / Module 02 - Curated AI Academic Research Tools

## Module 02 - Curated AI Academic Research Tools

### AI Academic Research Tools

For academic researchers, sifting through the overwhelming amount of research can be a challenge. To help with this, there are AI tools tailored to their needs. We have provided a brief list of the most relevant AI tools for the academic community.

These tools are invaluable in a world where academic knowledge is growing rapidly, making it easier for researchers to access, understand, and stay updated in their fields. Please submit any new suggestions!

TABLE OF CONTENTS
1  AI Academic Research Tools
2  General AI ChatBots
3  Chatbots Geared Towards Academia
4  Literature Review
5  Gauging Research Consensus
6  AI-Powered Flowchart Tool
7  References

### General AI ChatBots

General AI chatbots provide versatile assistance for various research tasks, from answering questions to offering research guidance.

- OpenAI ChatGPT

Figure 4: Page for module 02.

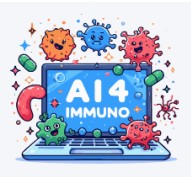

Q Search AI for Immunology

Modules / Module 03 - Using AI Chatbots in Research

## Module 03 - Using AI Chatbots in Research

### Prompt Engineering for Research Questions

For this module, we discuss the differences in AI chatbot performance and outputs when given the same prompt. We also share resources for scientists hoping to better understand and utilize prompt engineering principles for their own research questions.

TABLE OF CONTENTS
1  Prompt Engineering for Research Questions
2  Prompt Engineering Basics
3  Additional Prompts
4  Chatbot Comparison
5  Chatbot Performance Based on Four Metrics
6  Update to ChatGPT Gives it Eyes and Ears
7  Google's Bard
8  Microsoft's Bing Chat

### Prompt Engineering Basics

Prompt engineering refers to the practice of designing effective inputs that will result in the optimal outputs from generative AI tools. There are numerous tips online using prompt engineering techniques to help make AI chatbots work more effectively for your task.

Here is an example of a Reddit post on r/ChatGPT which claimed very lofty performance gains using this one prompt. Users said to copy and paste this prompt, keep providing details, and the prompt should continue to improve. Keep iterating until you craft the prompt you need.

Figure 5: Page for module 03.

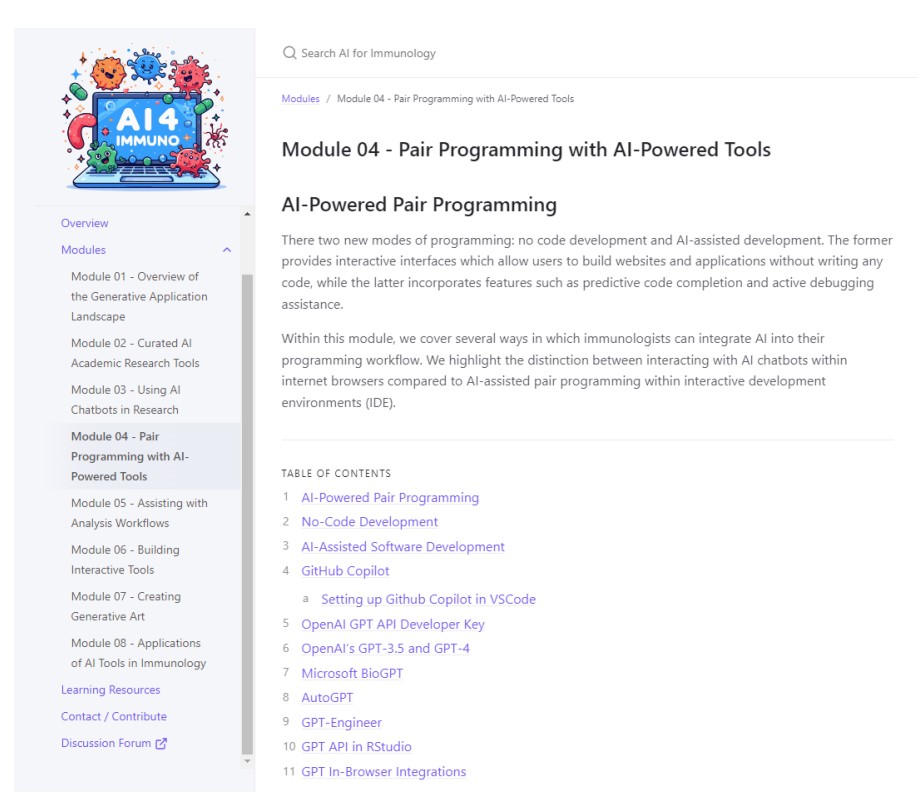

Figure 6: Page for module 04.

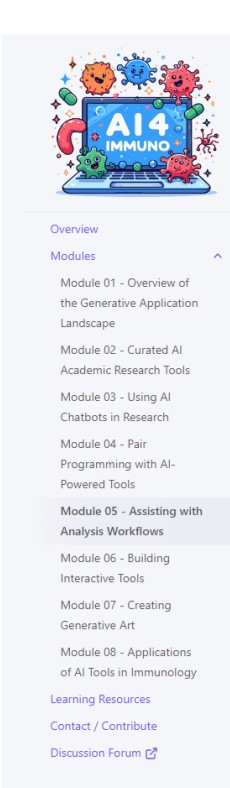

Q Search AI for Immunology

## Module 05 - Assisting with Analysis Workflows

### Generative AI for Data and Analytics

In this module, we focus on streamlining and simplifying the various stages of data analysis workflows. Standard preprocessing steps, often time-consuming, are addressed, including data cleaning, parsing, and creating high-level overviews. We provide resources which offer guidance on breaking down complex tasks into manageable steps, constructing effective workflows and roadmaps, and seamlessly connecting individual steps into a standard preprocessing pipeline. All of the above provide immunologists with practical considerations for enhancing their data analysis processes.

TABLE OF CONTENTS

### Data Preprocessing Tasks

ChatGPT can be a valuable integration into existing data science workflows. Data cleaning and preprocessing are often very time-consuming steps for any big analysis projects. Luckily, a tutorial on KDnuggets outlines how to use ChatGPT to help with a few tasks. Check out the site for step-by-step prompt and code examples.

- Fetch and load the dataset
- Check for missing values

Figure 7: Page for module 05.

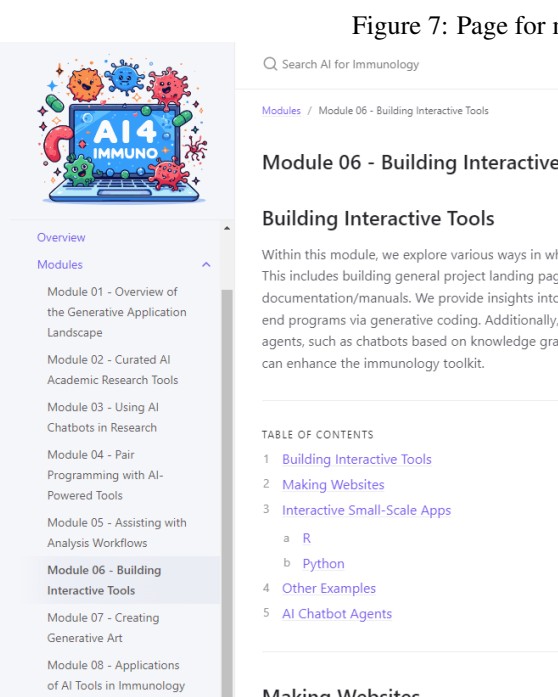

Q Search AI for Immunology

## Module 06 - Building Interactive Tools

### Building Interactive Tools

Within this module, we explore various ways in which immunologists can create interactive resources. This includes building general project landing pages, browser applications, and documentation/manuals. We provide insights into templating, and constructing both front and back-end programs via generative coding. Additionally, we delve into the realm of interactive web-based agents, such as chatbots based on knowledge graphs, offering a look at how these personalized tools can enhance the immunology toolkit.

TABLE OF CONTENTS

### Making Websites

How To Use Midjourney, AI Art, and ChatGPT to Create an Amazing Website A video outlining a process for creating websites using a combination of generative AI tools.

Figure 8: Page for module 06.

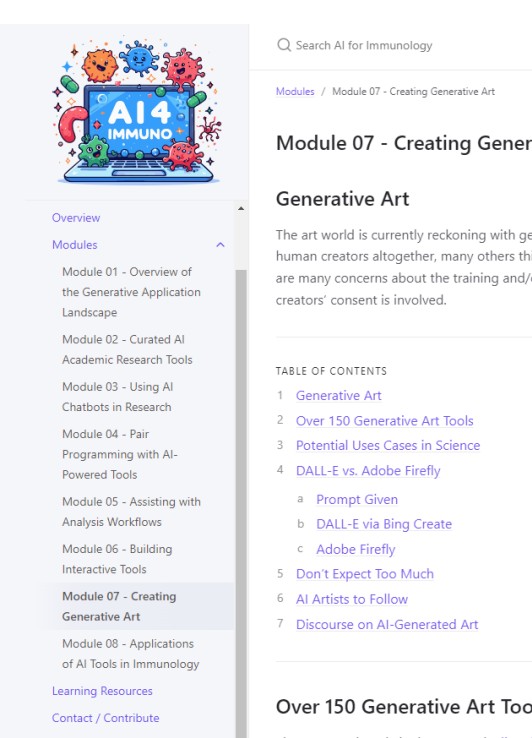

Search AI for Immunology

## Module 07 - Creating Generative Art

### Generative Art

The art world is currently reckoning with generative AI. While some individuals consider AI to replace human creators altogether, many others think of AI as an additional tool which aids in creation. There are many concerns about the training and/or input data used to power these models and whether creators' consent is involved.

### Over 150 Generative Art Tools

The OpenTools website has curated a list of over 150 generative art tools which are currently available. Each may have different pricing models or availability.

Figure 9: Page for module 07.

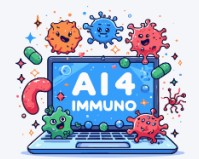

Search AI for Immunology

## Module 08 - Applications of AI Tools in Immunology

### Applications of AI Tools in Immunology

In this module, we feature examples of AI-powered applications in the field, such as an application where researchers trained an immunology knowledge assistant or generated an interactive web dashboard. We are actively gathering prime examples of AI-assisted development in immunology. Our aim is to highlight incredible projects at this exciting intersection.

Check back for more examples in immunology!

#### Jane the Immunology Knowledge Assistant

Here is an awesome example of an AI application in immunology called "Jane: the Immunology Knowledge Assistant". A large language model was fed Janeway's Immunobiology textbook plus the OMAPs from the Human Reference Atlas. We couldn't find the creator of this AI tool, but would love to credit them!

Figure 10: Page for module 08.

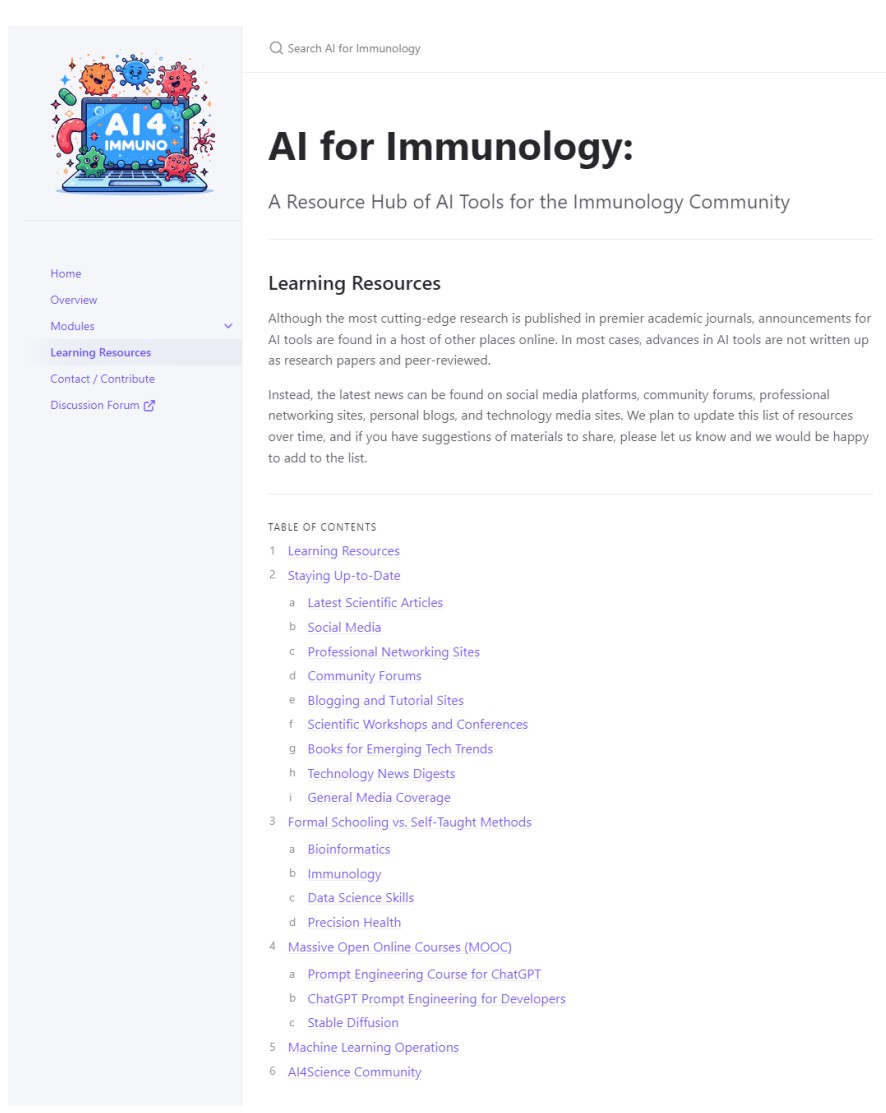

Figure 11: Page for learning resources.

