# OpenReview forum: "Immunology Meets Artificial Intelligence: Expanding Our Scientific Toolbox"
_NeurIPS.cc/2023/Workshop/AI4Science — NeurIPS2023-AI4Science Poster_

### Official Review · Reviewer_zGeT · 2023-10-22
**Serious concerns regarding contributions**

**Rating:** 3
**Confidence:** 4

**Review:**

While the proposed work presents a well-organized website serving as an AI resource hub with curated modules and resources, it raises questions about its novelty and research contribution. The primary concern is that it appears to be more of a compilation and resource curation effort rather than a research endeavor aimed at identifying and addressing specific research problems.
The paper's focus on describing a system may be better suited to a system description paper rather than a research paper, and such papers may fall outside the typical scope of acceptance.
I would suggest the authors consider how their work can make a more direct and novel contribution to the field of AI research, such as addressing specific research challenges, proposing new methodologies, or providing empirical results to support their claims.

---

### Meta-Review · Area_Chair_1631 · 2023-10-27

**Recommendation:** Reject
**Confidence:** 3

**Metareview:**

I am not an expert on immunity, yet I can see the merits of this paper. I agree with reviewer zGeT that this paper is not novel from the ML aspect. Meanwhile, it also means it can spark a lot of inspiring viewpoints from the ML aspects. Now, we have many tools utilizing group symmetry, quantum physics, statistics, and deep learning for solving fundamental problems, and this AI 4 Immuno opens a new venue.

My main concern: Though I can get the potential of AI 4 Immuno, the current draft is a little vague. What would be better to present/communicate to the ML community is to include the following:
- An abstraction of the problem that can be translated into more ML language, e.g., the conformation generation is to learn $p(x)$.
- Basic statistics of the datasets.
- More examples.